

# Duration of hemodialysis associated with cardio-respiratory dysfunction and breathlessness: a multicenter study

Kornanong Yuenyongchaiwat[1,2], Phatsara Vasinsarunkul[1], Phoomipat Phongsukree[1], Kodchaphan Chaturattanachaiyaporn[1] and Opas Tritanon[3]

[1] Physiotherapy Department, Faculty of Allied Health Sciences, Thammasat University, Pathumtani, Thailand
[2] Thammasat University Research Unit in Physical Therapy in Respiratory and Cardiovascular Systems, Thammasat University, Pathumthani, Thailand
[3] Division of Nephrology, Department of Internal medicine, Faculty of Medicine, Thammasat University, Pathumtani, Thailand

## ABSTRACT

**Background**. Patients with hemodialysis suffer with protein-energy wasting and uremic myopathy lead to lack of physical activity and poor functional performance. However, ventilation abnormality in patients undergone hemodialysis remains controversial regarding the respiratory impairment. Therefore, the study aimed to determine the effect of duration of dialysis on respiratory function.

**Methods**. A multicenter study with cross-sectional study was designed in four hemodialysis outpatient clinics. Respiratory muscle strength (i.e., maximal inspiratory pressure (MIP) and maximal expiratory pressure (MEP)) pulmonary function test (i.e., forced vital capacity (FVC), forced expiratory volume in one second ($FEV_1$) and $FEV_1$/FVC ratio), functional capacity (6-minute walk test) and sensation of breathlessness were assessed prior to dialysis.

**Results**. A total of 100 hemodialysis patients were recruited with 38 females and 62 males. An average of duration of hemodialysis was $5.93 \pm 4.96$ years. Decreased MIP values, $FEV_1$ values, FVC values, %$FEV_1$ and %FVC were noted in patients with long duration of dialysis (defined as $\geq 5$ years of dialysis) compared to those with short duration of dialysis ($p_s < .05$). In addition, increased sensation of breathlessness was observed in patients with long duration of dialysis ($p < .05$). Furthermore, participants with long duration of dialysis had an increased risk of ventilatory restriction (OR 6.093, $p = .007$).

## INTRODUCTION

Chronic renal failure (CRF) is not only impact on loss of renal function, it has been also presented in cardiorespiratory problems including shortness of breathing. The effect of muscle wasting and uremia in patients with CRF who are on hemodialysis resulting in the respiratory and peripheral muscle weakness due to impair oxygen uptake, consumption, transportation and decrease in protein synthesis (*Bark et al., 1988*).

Corresponding author
Kornanong Yuenyongchaiwat, ykornano@tu.ac.th

Pulmonary complication is one of the major problems in patients with CRF that is because of fluid overload, pulmonary calcification and fibrosis and pulmonary edema (*Craddock et al., 1977*). In addition, inflammation process and protein-energy wasting contributed to cardiovascular disease and these may develop pulmonary impairment in uremic patients (*Yoon, Choi & Yun, 2009*). A retrospective longitudinal study in Korea found that impairment of pulmonary function was associated with increase in the development of CRF (*Kim et al., 2018*). The prevalence of obstructive pulmonary function increased across GFR from 6% in GER-2 to 11% in GER-5 (*Mukai et al., 2018*). Compared to healthy individuals, patients with CRF shown decreased respiratory muscle strength and lung function (*Bark et al., 1988*; *Karacan et al., 2006*). Besides, a restrictive lung disease was found 36% of CRF patients with GFR <15 L/min/1.73 $m^2$ which markedly higher as GFR declined (*Murtagh, Addington-Hall & Higginson, 2007*). Previous study reported the prevalence of lung dysfunction was related to declining glomerular filtration rate (GFR) and the prevalence of dyspnea symptom was 35% among patients with end-stage renal disease (*Murtagh, Addington-Hall & Higginson, 2007*). In addition, patients with CRF receiving hemodialysis or patients with end stage renal disease (i.e., a GFR below 15 ml/min/1.73 $m^2$) are associated with high prevalence of cardiovascular disease e.g., hemodynamic stress, myocardial stress and injury (*Ahmadmehrabi & Wilson Tang, 2018*). Therefore, patients treated with hemodialysis are related to the risk of poor cardio-respiratory function. However, results of the relationships CRF patients and cardio-respiratory performance are quite inconsistent, with some suggesting obstructive and other restrictive pulmonary impairment in CRF. It might be differences in sample size, and participants may account for such differences. Therefore, the present study aimed to explore the relationships between cardio-respiratory performance in end stage renal failure patients undergoing hemodialysis.

## MATERIALS & METHODS

The study was approved from the Ethics Human committee of Thammasat University, based on Declaration of Helsinki, the Belmont report, CIOMS guidelines, and the International practice (ICH-GCP) COA No. 334/2560, and also the Ethics in Human Research Committee of the Thammasat Hospital Medicine. All participants were informed and gave written consent prior to the study. A total of 100 patients with end stage renal disease on hemodialysis were recruited in four hemodialysis centers. Those underwent hemodialysis at least three times per weeks and more than 3 months. Participants aged between 30–75 years old both males and females were recruited. Individuals who had history of no-smoking or non-current smoker or ex-smoker are included. A current smoker is defined as at least one cigarette per day within 1 week and ex-smoker is defined as participants who had ceased cigarette smoking in 6 months ago prior to the test (*Thornton, Lee & Fry, 1994*). The participant who had neurological problems (e.g., stroke), chronic cardio-respiratory disease (e.g., chronic cough, obstructive sleep apnea, history of chronic obstructive pulmonary disease or restrictive pulmonary disease, chronic heart failure), mental health problems, uncontrollable blood pressure (e.g., resting systolic blood
pressure over 200 mmHg or resting diastolic blood pressure more than 120 mmHg) were excluded.

Pulmonary function tests (i.e., forced vital capacity: FVC, and forced expiratory volume in one second: $FEV_1$) were assessed by the spirometeric meter (Carefusion MicroLab, United Kingdom) and a calibration was performed prior to the test. In addition, the respiratory muscle strength (inspiratory muscle and expiratory muscle strength) were assessed by respiratory pressure meter, which is a RPM01 (Micro Medical Ltd., United Kingdom). Individuals were asked to exhale slowly and completely and then inhale deeply and sustained pressure for 1.5 s to evaluate the maximal inspiratory pressure (MIP). Regarding to the maximal expiratory pressure (MEP), participants were instructed to exhale deeply and hold for 1.5 s. these individuals performed 3–5 MIP and MEP maneuvers, with the highest two within 10 $cmH_2O$ was recorded. These tests were followed by a recommendation of the *American Thoracic Society/European Respiratory Society (2002)*. Functional capacity is defined as 6-minute walk test (6MWT) and all individuals were asked to walk 30 m straight along a corridor in 6 min (*American Thoracic Society, 2002*). In addition, the participants were requested to rate the sensation of breathless during the past four weeks. Self-reported perceived breathlessness was measured using a numerical scale (1–5) with 1 corresponding to "very difficult to breath" and 5 corresponding to "not difficult to breath"; lower scores higher breathlessness. Furthermore, all participants were required to performed these tests prior to underwent hemodialysis. The determination of pulmonary ventilation, an obstructive lung disease is defined as FEV1/FVC < 0.70 and restrictive impairment is defined as $FEV_1$/FVC $\geq$ 0.70, and %FVC <80 (*Johnson & Theurer, 2014*).

Data were displayed as mean and standard deviation, percentage, as appropriate. Statistical significance was set as the level of $p < .05$. Normality was verified using the Komogorov Siminov Goodness of Fitness test. A chi-square test was used to analyze differences between short and long duration of dialysis and type of respiratory function. Pearson correlation was conducted to determine whether the duration of hemodialysis was related to cardio-respiratory function and sensation of breathlessness. To examine association between these relationships after controlling for age and sex, partial correlation analysis was conducted. Comparisons between two groups by categorized dialysis duration (defined as <5 years and $\geq$5 years) were assessed with unpaired $t$-test.

## RESULTS

The mean age of patients with end stage renal disease was 51.54 ± 11.19 years. A total of 100 participants with 38 females and 62 males completed pulmonary function test respiratory muscle strength testing and 6MWT. A mean duration of hemodialysis was 5.93 ± 4.96 years. According to categorize the lung function into three groups; normal, obstructive and restrictive lung impairment, the result revealed that only 17 individuals (17.00%) were categorised as a normal pulmonary function, whereas 82 participants (82.00%) were categorised as a restrictive ventilatory impairment, none of participants was defined as obstructive lung function. Table 1 displays the descriptive data of CRF patients.

**Table 1  Demographic and characteristics of chronic renal failure patients.**

|  | N (%) | mean ± SD |
|---|---|---|
| Sex |  |  |
|     Female | 38(38.00) |  |
|     Male | 62(62.00) |  |
| Age |  | 51.54 ± 11.19 |
| Duration of hemodialysis (yr) |  | 5.93 ± 4.96 |
| 6-MWD (meters) |  | 373.04 ± 107.80 |
| MIP (cmH$_2$O) |  | 61.42 ± 28.51 |
| MEP(cmH$_2$O) |  | 67.64 ± 30.23 |
| FVC (L) |  | 2.09 ± 0.74 |
| FVC (%) |  | 64.71 ± 18.12 |
| FEV$_1$(L) |  | 1.94 ± 0.67 |
| FEV$_1$(%) |  | 71.03 ± 20.87 |
| FEV$_1$/FVC |  | 94.12 ± 5.84 |
| PEFR (L/min) |  | 273.94 ± 122.66 |
| PEFR (%) |  | 58.68 ± 23.05 |
| Sensation of breathlessness |  | 3.76 ± 1.15 |

**Notes.**

MIP, maximal inspiratory pressure; MEP, maximal expiratory pressure; FVC, forced vital capacity; FEV$_1$, forced expiratory volume in the first second; 6MWD, 6-minute walk distance; PEFR, peak expiratory flow rate.

Bivariate correlations between duration of hemodialysis and pulmonary function, respiratory muscle strength tests and 6MWT were presented in Table 2. Duration of hemodialysis was negatively associated with FVC values, FEV$_1$ values, MIP values, %FVC, % FEV$_1$, 6-minute walk distance (6MWD) and sensation of breathlessness ($p <$ .05). In addition, these relationships remained after adjusting for age and sex; individuals with long duration of hemodialysis still had a poor cardio-respiratory performance (i.e., pulmonary function, respiratory muscle strength and functional capacity) and high sensation of breathlessness.

Several studies have been proposed for choosing different cutpoints of short-term and long-term duration of hemodialysis. For example, *Hou et al. (2014)* defined a long-term of hemodialysis as at least one year of hemodialysis. Another study defined as an average 51 months for the long period of hemodialysis treatment (*Chazot et al., 2001*). Here, the study is considering an equal number of the participants. Therefore, the categorization of period of hemodialysis was presented using cut point of five years. The study revealed that 50 out of 100 (50.00%) of the CRF patients were categorized as long duration of dialysis ($\geq$5 years). Individuals with short duration of dialysis had higher FVC values, %FVC, FEV$_1$ values, %FEV$_1$ MIP values and also lower sensation of breathlessness than those long duration of dialysis (see Table 3). Individuals with long duration of hemodialysis (defined as a duration of hemodialysis $\geq$ 5 years) had shorter duration of 6MWT than those with short duration of hemodialysis; however, these did not reach the conventional criterial ($p >$ .05). Patients with CRF who had a long duration of hemodialysis had a high

Yuenyongchaiwat et al. (2020), *PeerJ*, DOI 10.7717/peerj.10333

**Table 2  Correlation between duration of hemodialysis and respiratory function in chronic renal failure patients.**

| | MIP (*p*-value) | MEP (*p*-value) | 6-MWD (*p*-value) | FVC (*p*-value) | %FVC (*p*-value) | FEV1 (*p*-value) | %FEV1 (*p*-value) | FEV1/FVC (*p*-value) | PEFR (*p*-value) | %PEFR (*p*-value) | Sensation of Breathlessness (*p*-value) |
|---|---|---|---|---|---|---|---|---|---|---|---|
| HD | −0.2.75 (0.006) | −0.187 (0.063) | −.210 (0.036) | −0.296 (0.003) | −0.248 (0.013) | −0.307 (0.002) | −0.255 (0.011) | −0.025 (0.805) | −0.072 (0.478) | 0.019 (0.848) | −0.220 (0.028) |
| HD[a] | −0.229 (0.024) | −0.135 (0.186) | −0.200 (0.048) | −0.265 (0.008) | −0.238 (0.018) | −0.279 (0.005) | −0.232 (0.021) | −0.031 (0.759) | −0.007 (0.947) | 0.032 (0.756) | −0.224 (0.027) |

**Notes.**

[a]Controlling for age and sex.

HD, hemodialysis; FVC, forced vital capacity; FEV1, forced expiratory volume in the first second; PEFR, peak expiratory flow rate; MIP, maximal inspiratory pressure; MEP, maximal expiratory pressure.

**Table 3 Displays the differences in pulmonary function and respiratory muscle strength by categorized duration of hemodialysis.**

| | Duration of HD < 5yrs ($n = 50$) | Duration of HD ≥ 5years ($n = 50$) | t(98) | p-value | 95% CI | Effect size Cohen's d |
|---|---|---|---|---|---|---|
| MIP (cmH$_2$O) | 67.42 ± 25.47 | 55.42 ± 30.33 | 2.142 | 0.035 | 0.88 to 23.12 | 0.428 |
| MEP(cmH$_2$O) | 71.10 ± 25.93 | 64.18 ± 33.91 | 1.146 | 0.254 | −5.06 to 18.90 | 0.229 |
| 6-MWD (meters) | 386.14 ± 122.29 | 359.94 ± 90.39 | 1.218 | 0.226 | −16.48 to 68.88 | 0.244 |
| FVC (L) | 2.29 ± 0.74 | 1.89 ± 0.69 | 2.777 | 0.007 | 0.11 to 0.68 | 0.559 |
| FVC(%) | 69.74 ± 18.27 | 59.68 ± 16.68 | 2.877 | 0.005 | 3.12 to 17.01 | 0.575 |
| FEV$_1$(L) | 2.12 ± 0.66 | 1.77 ± 0.63 | 2.706 | 0.008 | 0.09 to 0.60 | 0.542 |
| FEV$_1$(%) | 76.60 ± 21.56 | 65.46 ± 18.74 | 2.758 | 0.007 | 3.13 to 19.16 | 0.552 |
| FEV$_1$/FVC | 93.70 ± 5.78 | 94.54 ± 5.93 | −0.717 | 0.475 | −3.16 to 1.48 | 0.143 |
| PEFR (L/min) | 278.48 ± 126.02 | 269.40 ± 120.31 | 0.369 | 0.713 | −39.82 to 57.98 | 0.074 |
| (%) PEFR | 58.80 ± 24.63 | 58.57 ± 21.60 | 0.050 | 0.960 | −8.96 to 9.43 | 0.010 |
| Sensation of breathlessness | 4.08 ± 0.90 | 3.44 ± 1.28 | 2.892 | 0.005 | 0.20 to 1.08 | 0.578 |

**Notes.**

HD, hemodialysis; MIP, maximal inspiratory pressure; MEP, maximal expiratory pressure; FVC, forced vital capacity; FEV1, forced expiratory volume in the first second; 6MWD, 6-minute walk distance.

prevalence of restrictive pulmonary impairment at 47% ($n = 47$) compared to individuals with short duration of dialysis (3.00%, $n = 3$) ($\chi^2 = 8.575$, $p = .003$).

# DISCUSSION

The present study examined the relationship between duration of hemodialysis and cardio-respiratory performance in patients with CRF. The study found the association of MIP values, FVC values, predicted FVC, FEV$_1$ values, predicted FEV$_1$, 6MWT and the sensation of breathlessness. The study also shown that compared with short duration of hemodialysis (defined as <5 years), patients who had a long duration of hemodialysis displayed a reduction in of MIP values, FVC values, predicted FVC, FEV$_1$ values, predicted FEV$_1$, 6MWT and increased in the sensation of breathlessness.

The patients with hemodialysis in the study had low mean MIP values (61.42 ± 28.51 cmH$_2$O); indicating inspiratory muscle weakness (defined as the ATS/ERS guideline which MIP values of <80 cmH$_2$O). Decreased respiratory muscle value was consistent with other studies (*Bark et al., 1988*; *Fassbinder et al., 2015*; *Karacan et al., 2006*). *Karacan et al. (2006)* reported inspiratory muscle was displayed 66.5 ± 23.4 cmH$_2$O in 27 hemodialysis patients. In addition, *Bark et al. (1988)* found the MIP was 58.2 ± 24.9 cmH$_2$O in 10 patients with CRF group. Therefore, the respiratory muscle strength weakness was noted in patients with CRF on hemodialysis. This might be due to uremic myopathy that leads to reduced strength in skeletal muscles and diaphragm (*Tarasuik, Heimer & Bark, 1992*). *Tarasuik, Heimer & Bark (1992)* reported that force and frequency of diaphragmatic muscle was decreased by 15% in the moderate uremia and by 45% in the severely uremic rats. In addition, they found the fatigability of diaphragm was increased in the moderately and severely uremic rats (*Tarasuik, Heimer & Bark, 1992*). Therefore, muscle-related CRF complication may be explained by uremic myopathy which can be attributed structural changes (e.g.,

a decrease in excitability-contractility coupling of respiratory muscle) (*Bark et al., 1988*). Further, a deficit of vitamin D, anemia, hypophosphatemia, and malnutrition have been reported in patients with CRF (*Karacan et al., 2006*). One mechanism linking uremia to muscle weakness is altered active Calcium-transport of sarcoplasmic reticulum. Heimberg et al. found that the Calcium-transport system in uremic rabbits was changed; decrease in the calcium influx rate, increase in the calcium permeability and resulting decreased concentrating ability of sarcoplasmic reticulum (*Heimberg et al., 1976*). Those might be, at least in part, due to resulting in a reduction of respiratory muscle strength.

In the study, decrease in pulmonary function has been observed in patients receiving hemodialysis. Decreased spirometry parameters in patients undergoing hemodialysis was reported in several studies (*Kovacevic et al., 2011*; *Kovelis et al., 2008*). These CRF patients on hemodialysis showed a restrictive ventilatory defect (defined as FVC <80% and $FEV_1/FVC \geq 0.7$) with have been similarly reported from the previous studies that is restrictive pulmonary impairment was common complication in patients with advanced CRF (*Karacan et al., 2006*; *Kovelis et al., 2008*). This is due to fluid overload, increasing interstitial oedema and bronchial wall decongestion resulting in decreased pulmonary function (*Kovelis et al., 2008*; *Yilmaz et al., 2016*). In addition, uremia has been reported to related pulmonary microcirculation dysfunction (*Ewert et al., 2002*). Another is that the decrease in pulmonary compliance of the thoracic wall that might be a muscle wasting with protein-energy wasting and inflammation (*Mukai et al., 2018*). Further, Bark et al. found that decreased respiratory muscle strength was related to a decrease in vital capacity among patients with CRF (*Bark et al., 1988*). Therefore, impaired respiratory muscles strength and poor ventilator capacity or lung restriction were noted in patients with end stage renal disease on hemodialysis. Previous studies have been reported that the relationships between lung function and inflammation were negatively associations (*Bolton et al., 2011*; *Hancox et al., 2016*). It is possible that a restrictive lung function may be in part, increased systematic inflammation. Therefore, cardio-pulmonary-renal interactions have been reported in CRF can cause a respiratory restriction, impaired gas exchange and also decreased exercise capacity (*Husain-Syed et al., 2015*). However, further studies need to explore the relationship with impaired cardiopulmonary function in CRF patients.

According to shortness of breathing, it was found that duration of hemodialysis was negatively associated with breathlessness even after controlling for age and sex; longer duration of hemodialysis higher breathlessness in hemodialysis patients. The prevalence of dyspnea was also reported between 20% and 60%. *Palamidas et al. (2014)* found that 100% of hemodialysis patients had displayed mild to moderated degree of chronic dyspnea (measured by Modified Medical Research Council Dyspnea Scale) before hemodialysis. However, it should be noted that only 25 hemodialysis patients were included in the study. In addition, Palamidas et al. assumed that an accumulation of excess lung water and pulmonary edema in patients with hemodialysis leads to premature airway closure and gas trapping. Further, a high ventilatory drive appealed before the

hemodialysis which reflects increased respiratory effort and work of breathing (*Palamidas et al., 2014*). Therefore, impaired respiratory muscle and ventilatory impairment might be perceived as shortness of breathing from the hemodialysis patients.

The reduction in functional capacity determined by the 6MWT has been observed in several studies (*Barril et al., 2018*; *Fassbinder et al., 2015*). *Fassbinder et al. (2015)* found the distance of 6MWT was $418.67 \pm 117.3$ m in 27 hemodialysis patients with an average age was $58.15 \pm 10.84$ years old. *Barril et al. (2018)* found that the walking distance was $586.39 \pm 155$ m in 108 patients with advanced CRF, mean aged was $67.36 \pm 12.97$ years which was lower than the predicted values. Further, they reported the GFR was associated with the distance of 6MWT in patients with advanced CRF (*Barril et al., 2018*). In the present study, the distance of 6MWT was $373.04 \pm 107.80$ m with a mean overall age of the participants was $51.54 \pm 11.19$ years and duration of hemodialysis was $5.93 \pm 4.96$ years. Thus, the impairment in functional capacity in the present study show a lower distance compared to the previous studies (*Barril et al., 2018*; *Fassbinder et al., 2015*). It has been already known that the demographic data such as age height, body weight and race are associated with distance of 6MWT (*American Thoracic Society, 2002*; *Poh et al., 2006*). In addition, the duration of hemodialysis might be in part of the walking distance in patients with CRF. Therefore, a prospective cohort study should be explored.

A number of limitation should be a consideration. A cross sectional study was designed which could not be a causal relationship. A confounding factors such as volume overload, laboratory analysis of renal function, and comorbidity was also not recorded. Other investigations such as laboratory investigation (e.g., arterial blood gas) or chest X-ray (e.g., interstitial lung disease) were not reported. In addition, all participants were only clinically stable; therefore, the results might not have generalized to CRF population as a whole. Finally, the sample size of the study was not calculated; however, the results of the study have reanalyzed with statistical power to detect the effect. According to a total number of 100 participants, the effect size for FVC values was 0.56; therefore, the retrospective statistical power was 0.79 for a 2-tailed alpha was 0.05. Thus, the findings of study are commensurate adequate statistical power. However, the future study need to consider regarding the sample size calculation.

## CONCLUSIONS

Patients with long duration of hemodialysis displayed a reduction of respiratory function, functional capacity and also increased breathlessness. In addition, these individuals shown a risk of restrictive ventilatory impairment.

## ACKNOWLEDGEMENTS

The authors would like to thank all staff from hemodialysis Center at Thammasat University Hospital, Karunvej Hospital, Pat-Rangsit Hospital, and Phatara-Thonburi Hospital, for their kind support and their help. In addition, we would also like to thank the participants and their caregivers for participating in the study.

### Funding
This work was supported by Thammasat University Research Unit in Physical Therapy in Respiratory and Cardiovascular Systems. The funders had no role in study design, data collection and analysis, decision to publish, or preparation of the manuscript.

### Grant Disclosures
The following grant information was disclosed by the authors:
Thammasat University Research Unit in Physical Therapy in Respiratory and Cardiovascular Systems.

### Competing Interests
The authors declare there are no competing interests.

### Author Contributions
- Kornanong Yuenyongchaiwat conceived and designed the experiments, analyzed the data, prepared figures and/or tables, authored or reviewed drafts of the paper, and approved the final draft.
- Phatsara Vasinsarunkul, Phoomipat Phongsukree and Kodchaphan Chaturattanachaiya-porn performed the experiments, prepared figures and/or tables, and approved the final draft.
- Opas Tritanon conceived and designed the experiments, analyzed the data, authored or reviewed drafts of the paper, and approved the final draft.

### Human Ethics
The following information was supplied relating to ethical approvals (i.e., approving body and any reference numbers):

The Ethics Human committee of Thammasat University and the Ethics in Human Research Committee of the Thammasat Hospital Medicine approved the study (COA No. 334/2560).

### Data Availability
Raw data is available as a Supplemental File.

### Supplemental Information
Supplemental information for this article can be found online at http://dx.doi.org/10.7717/peerj.10333#supplemental-information.

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
