# Peer review of "Duration of hemodialysis associated with cardio-respiratory dysfunction and breathlessness: a multicenter study"

_PeerJ, doi:10.7717/peerj.10333_

## Round 0.1 · original submission · Major Revisions

Address all comments and issues raised during the review process clearly and unequivocally.

Reviewer 1 ·

Basic reporting

Writing generally good; needs slight English language polishing which will likely be done in the editing process.

In the abstract, the sentence "These patients are also related to poor respiratory performance due to pulmonary impairment" should be removed, as it is presumptively asserting facts which this study is aiming to prove.

Experimental design

It should be stated explicitly in the abstract and in the methods that this was a prospective study.

There needs to be more detail about what "chronic cardio-respiratory conditions" were exclusion criteria; does this include any restrictive or obstructive respiratory pathology? Obstructive sleep apnoea? Heart failure? How about smoking history?

How was duration of haemodialysis defined for the analysis- in years, or months?

Where continuous data was expressed as mean and standard deviation, were tests of normality performed to check if this data was normally distributed? (If any of the continuous data variables were non-normally distributed, median +/- IQR would be more appropriate)

Validity of the findings

Line 142 to 144 do not make sense: "Patients with CRF who had a long duration of HD had a high prevalence of restrictive pulmonary impairment at 47% (n = 47) compared to individuals with normal pulmonary function (3.00%, n = 3) (X2 = 8.575, p = .003)." This paragraph is concerned with comparisons between long duration of HD and short duration of HD groups, not Chi-Squared comparisons within the same group.

Table 4 is totally missing. Furthermore, even if Table 4 were included, it is unclear what is the purpose of the logistic regression analysis. Is it univariate or multivariate logistic regression analysis?

It appears that the binary dependent variables being studied in the regression are long duration of HD and short duration of HD. However, these are not outcomes that are influenced or predicted by other independent variables- they are simply a function of the passage of time since the patient started dialysis. Using clinical/respiratory parameters to "predict" whether someone has been on dialysis for a long time has no practical utility. I would remove the logistic regression analysis altogether.

In the discussion, line 196/197 "The present study found that the association between FVC and inspiratory muscle strength was observed in HD (data was not shown)" is inadequate: if this is an interesting point, this data should be shown in the manuscript with the appropriate statistical analysis. Perhaps this data, comparing the assocation between different respiratory variables in this cohort, should be presented instead of the above-mentioned logistic regression.

Reviewer 2 ·

Basic reporting

- The number of literature references could be more extensive to provide suitable context.

- The article structure is suitable however in this topics below I suggest:
Introduction: Hypotheses should be described in this section. What is the rational for including the comments about GFR and this variable was not described in methods, tables or discussion? I think that GFR is a very important variable.

Experimental design

The research question was well defined and relevant however there are many studies about these questions. Rigorous investigation of technical and ethical aspects.

The methods section was sufficient but I have a question: Why the authors did not use reference formulas to predict distance in TC6 and values of PImax and PEmax? These equations are essential to described and interpreted because comparing values against healthy adults.

Was the sample size estimated?
Why the authors chose the categorized dialysis duration (defined as < 5years and ≥5
110 years)? Was based in previous literature?

Validity of the findings

The results showed a benefit to literature however, the external validity is hard to be concluded. The power of study was not showed without a priori calculate of sample size.

Information about laboratory analysis of renal function would be very important in this population.

Conclusions were linked to original questions.

Additional comments

General comments: Thank you very much for allowing me to review this manuscript.
Overall, this article is very interesting and does provide additive data to the existing published literature.

---

## Round 0.2 · accepted · Accept

Both referees found your manuscript substantially improved. All issues raised were satisfactorily addressed. The editor concurs

Reviewer 1 ·

Basic reporting

No further issues

Experimental design

No further issues

Validity of the findings

No further issues

Reviewer 2 ·

Basic reporting

The changes done were adequate.

Experimental design

The changes done were adequate.

Validity of the findings

The changes done were adequate.

Additional comments

The changes done were adequate.